# Do the Elderly Need Wider Parking Spaces? Evidence from Experimental and Questionnaire Surveys

**Weite Lu** *[ID]**, Chunqin Zhang, Xunyou Ni and Haiqiang Liu**

School of Civil and Architectural Engineering, Zhejiang Sci-Tech University, Hangzhou 310018, China; cqzhang@zstu.edu.cn (C.Z.); nixunyou@zstu.edu.cn (X.N.); liu.haiqiang@zstu.edu.cn (H.L.)
* Correspondence: weitelu@zstu.edu.cn; Tel.: +86-13706515892

**Abstract:** An excellent parking system can affect the willingness of the elderly to drive an automobile for travel and for participating in social activities. However, few studies have examined the requirement of width of current parking spaces for the elderly and which factors influenced the selection of wider or narrower parking space by older participants. Two studies have been carried out in order to fill gaps for these issues. The first one examined minimum width by having 130 individuals aged 60+ alight into parking spaces of different widths. The results showed that most older individuals needed wider-than-standard parking spaces. Some potential demographic factors were also examined by one-way analyses of variance. The second study was conducted to estimate the factors affecting selection of parking spaces that were wider or narrower than the standard. Based on analysis of data by a logistic regression model, the result presented that the selection was mainly affected by age, types of aids, driving experience, body mass index (BMI) and health condition. Finally, we propose a new concept of parking system, which will help older people with different needs to park safely and smoothly. These studies will promote the ability of governments to design more effective parking spaces to enhance freedom for older adults.

**Keywords:** the elderly; parking; width; one-way analyses of variance; logistic regression model

## 1. Introduction

Aging brings many challenges in the sustainable society [1,2]. Study on mobility of the elderly has shown that travel patterns of older adults tend to decrease due to deterioration in health and consequent reduced access to transportation [3–5]. However using public transport was reported to be a good option for the elderly. By using public transport, they do not endanger themselves or others while on the road or when parking. Nonetheless, some literature showed that few elderly reported problems with driving, but larger proportions of the elderly experienced problems in using public transport [4,6]. The elderly passengers may face impairments due to arthritis or problems in joints and face the risk of slipping and falling down when they are boarding and alighting the bus. In addition, although public transport systems in urban cities support and help the elderly to travel and move, driving a car is important for many older individuals to maintain their independence in suburbs or small cities because these areas do not provide adequate public transport systems. Both driving (or taking) a car and health conditions have been proven to be two main factors that play a significant role in mobility for the elderly in suburban areas [7]. In the West (e.g., the UK), the car is the most used mode of transport among older people in terms of distance travelled and trip frequency [8].

Much of the current literature has focused on how to improve bus service and physical design of vehicles to encourage elderly passengers to travel by bus [9,10], and also much research has been done

on driving behavior by the elderly [11,12], but few have reported their associated car parking behavior and the boarding or alighting behavior of the elderly. Aging may cause difficulty in the parking (or getting on or off) process because of many factors related to physical and cognitive functions [13]. Studying the appropriate width of parking spaces is also very important because a large number of the elderly are using cars (both as drivers and passengers) to travel.

Increasing the width of parking spaces for the elderly may help these individuals park and board or alight more easily [14]. However, there is little consensus around the world in regards to policies that widen parking spaces for the elderly. In most western countries, the width of parking spaces for the elderly is not specified by legislation and regulation. At the moment, people with disabilities are eligible for disabled parking permits in America although there are ongoing discussions in some jurisdictions about enlarging spaces to other people who have difficulty with travel, such as the elderly. However, in Asia, Japan and South Korea made special parking permits for the elderly and agreed that they can use 3.50 m-wide disabled parking spaces to make parking more convenient [15]. Japan even added some reserved parking spaces in public spaces that are 2.50–2.70 m-wide (the width of a standard parking space in Japan is 2.50 m) particularly for the elderly and for pregnant women [14]. It was also reported that some residential areas in China have set aside some wider-than-standard parking spaces for the elderly. However, China did not specify the value of width of these reserved spaces [16].

To the authors' knowledge, there is scant literature about the importance of wider parking spaces for older adults in Japan or in South Korea. In addition, there is little justification on how to determine the width of parking space for the elderly in these East Asian countries. As a whole, only Japan has minimally described that the width of reserved parking spaces for the elderly should be 20 cm wider than regular spaces [17]. Nevertheless, they did not carry out any investigation to see if the value was meaningful or not. Moreover, there was also a lack of data indicating that any local government considered the influential factors on the use of different sizes of parking spaces. Only Lu et al. [18,19] mentioned that body sizes and physical strength of individuals might be crucial influential factors.

Although there is a shortage of literature about testing of minimum width for the elderly, a few studies have pointed out standard parking spaces might be too narrow for the elderly, even though many elderly are using standard parking spaces. Kiyota et al. [14] invited ten participants over 60 years old to measure the width requirement of parking in Japan. The results indicated that the current 2.50 m-wide standard parking space is undersized for these participants. Lu et al. [20] enlarged the sample of elderly drivers (N = 5) who were using canes to continue the parking experiment. Their findings were in line with Kiyota et al.'s viewpoints and considered that 2.75 m should be more suitable for the elderly. Lu et al. [21] observed 25 older individuals who owned the formal parking permits specifically for the elderly and how they parked in 2.70 m-wide reserved parking spaces in a supermarket parking lot in Japan. They found that most older individuals were able to aboard and alight in these spaces smoothly.

To summarize, previous studies have indicated that the width of standard parking spaces may be inappropriate for the elderly in Asian countries. Nevertheless, some problems have not been completely resolved. First, the limited sample size cannot provide strong evidence to show the minimum width required by the elderly. Second, there are few studies about usage of parking space for the elderly in China. The data about parking width for the elderly in Japan may not be applicable in China. Third, there are few studies on which factors are related to the selection of wider or narrower parking space by the elderly. Quantifying behavior of wider or narrower parking space selection and identifying the correlation of use of parking spaces by the elderly is the first important step to understand and improve the parking system for older adults.

In this paper, we address gaps in the literature with two studies. The first one is an exploratory survey (in Section 2) to inspect the minimum parking width for different categories of the elderly. The findings of this study will test the hypothesis of whether the elderly require wider-than-standard parking spaces as well as provide some influential factors such as demographic factors. These influential factors will be used for a part of the questionnaire items and will be verified in the second study.

The second study is a questionnaire survey (in Section 3) to identify factors that are related to the selection of wider or narrower parking spaces. The potential influential factors including sociodemographic variables, transport variables and health-related variables are examined using a logistic regression model. Finally, in combination with the concept of sustainable transportation, we propose some policy adjustments for the current parking system, which includes a drop-off zone for buses' elderly passengers. We hope that the accessibility for elderly can be improved according to this study.

## 2. Experimental Survey

### 2.1. Method

#### 2.1.1. Data Sources

The research object of this study is elderly people aged 60 years and older living in the metropolis of Hangzhou, which is the capital city of Zhejiang Province and located in the eastern region of China. There are 1,740,000 people aged 60 years and older living in Hangzhou (22.53% of the total population), which shows that the percentile of the aged in this area is high [22]. One hundred and thirty older participants (aged 60–75 years) were randomly recruited from local car clubs and the Geriatric Service Association of Hangzhou. It is noteworthy that it is very difficult to find participants older than 75 years due to the fact that many older people have lost valid driving licenses. The participant group included 68 males (52.3%) and 62 females (47.7%). Twenty-seven were cane users and the remainder did not use any aid devices. The survey gathered information on participants' gender, age, body traits, a self-reported health condition (good/neither good nor bad/bad) and driving experience. This study was approved by the Geriatric Service Association of Hangzhou.

#### 2.1.2. Design

The main purpose of the survey was to ask participants to report the level of difficulty of parking, alighting or boarding in an assigned space.

Two changeable width parking spaces were designed by using 7.5 cm-wide moveable white flexible tape. The width of tape was not especially stated; however, parking space lines are typically 7.5 cm wide in China. Two parking spaces (spaces a and b), shown in Figure 1, were permanently set with a width of 2.50 m and a length of 5.00 m, which is the standard dimension of parking spaces in China [23]. To begin with, the alighting space between the two parking spaces was 1.10 m wide. The width of assigned space for the experiment was 3.60 m (i.e., the alighting space = 1.10 m plus the parking space a = 2.50 m), which equates to a standard disabled parking space in China. The experiment began with 3.60 m because we considered some participants might expect the space to be as wide as a standard disabled parking space.

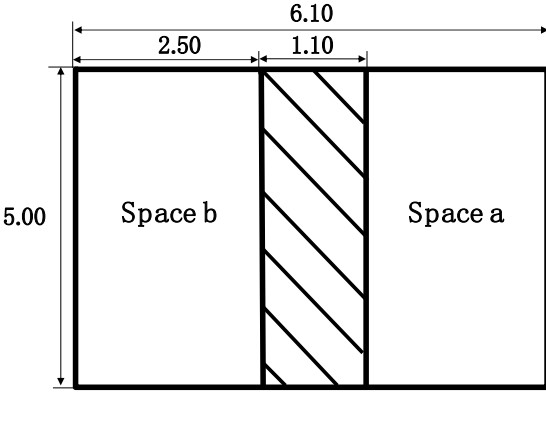

**Figure 1.** Set-up of the experimental parking area.

The experimental vehicles chosen were a 1.83 m-wide sedan and a 1.85 m-wide sedan without any automatic parking assistance, which are large-sized sedans in China [23]. We did not use smaller-sized vehicles in the study because we considered that if individuals can alight with a larger size car, the other smaller sized cars should give ample clearance and be acceptable. Additionally, all participants were asked to board and alight by themselves without any assistance.

The experimental survey was implemented on 4, 5 and 6 May, 2017, with fair weather.

### 2.1.3. Process

The main process in the experiment was:

(1) The 1.85 m-wide sedan was permanently parked in space b. In the survey, the sedan was parked in the middle of space b in order to avoid influence from other adjacent parked vehicles in a real-life situation [24].

(2) Participants drove the 1.83 m-wide sedan and parked in space a. Participants then alighted in the entire area of the space (i.e., space a plus alighting space). All participants reported whether the width was adequate to park or alight. In order to obtain data of minimum width, participants were asked to try to alight within the assigned width even if they expected they would not be able to alight successfully. The multiple choice responses to the difficulty of alighting on the form were: (a) I can alight without any problems; (b) I can alight but feel it is a little narrow; (c) I feel it is difficult to alight but I could do it; and (d) I could not alight. It is noteworthy that all participants were required to park nose-in in order to use the alighting space on the driver's side (note: right-hand driving in China).

(3) The sedan in space a was shifted. The space between a and b was consistently decreased at increments of 0.10 m by adjusting the white flexible tape.

(4) The experiment repeated step (2) and (3) until participants reported the parking space made it impossible to get off the car (i.e., answer (d) in the form).

Between two consecutive procedures, participants would take a two- or three-minute break because we believed fatigue occurred and accumulated while individuals repeated parking and boarding or alighting steps.

It was considered that the alighting and boarding time could be used to test the accuracy of self-reported data [19]. We respectively recorded the alighting and boarding time of each participant. However, the time recorded did not align with the self-reported conditions of our participants' bodies; i.e., the questionnaire responses from participants did not correspond with time needed to alight and board.

### 2.2. Results

The sample profile is presented in Table 1. About 90% of participants were 60–70 years old. The body mass index (BMI) of most participants was 18.5–25 (normal range). Close to two thirds of the participants claimed they were in good body condition. Most participants had 10+ years of driving experience.

**Table 1.** Characteristics of the participants (n = 130).

| Items | Participants | |
|---|---|---|
| | Count | Percentage (%) |
| Gender | | |
| Male | 68 | 52.3 |
| Female | 62 | 47.7 |
| Age (years) | | |
| 60–65 | 79 | 60.8 |
| 66–70 | 38 | 29.2 |
| >71 | 13 | 10.0 |
| Aids | | |
| Cane | 27 | 20.8 |
| None | 103 | 79.2 |
| Body weight (kg) | | |
| <40 | 4 | 3.1 |
| 40–<50 | 21 | 16.1 |
| 50–<60 | 51 | 39.2 |
| 60–<70 | 43 | 33.1 |
| ≥70 | 11 | 8.5 |
| Body height (cm) | | |
| <150 | 5 | 3.8 |
| 150–<160 | 22 | 16.9 |
| 160–<170 | 47 | 36.2 |
| 170–<180 | 51 | 39.3 |
| ≥180 | 5 | 3.8 |
| BMI | | |
| <18.5 | 11 | 8.5 |
| 18.5–<25 | 114 | 87.7 |
| 25–<30 | 5 | 3.8 |
| ≥30 | 0 | 0 |
| Health condition | | |
| Good | 83 | 63.8 |
| Neither good nor bad | 42 | 32.3 |
| Bad | 5 | 3.9 |
| Driving experience (years) | | |
| 0–<3 | 4 | 3.1 |
| 3–<10 | 21 | 16.2 |
| ≥10 | 105 | 80.7 |

### 2.2.1. Responses

Responses to different parking widths are shown in Table 2. The result shows that the width of 2.50–2.70 m was insufficient for all participants. About 72.3% of users (N = 94) confirmed they could alight at a width of 2.80 m, although the other 27.7% (N = 36) of users selected "I feel it is hard to alight but I could do it". All participants agreed that the width of 2.90 m was sufficient to alight. The gray cells in Table 2 highlight the possible minimum width values for the two groups.

**Table 2.** Responses of older participants in the experimental survey.

| Width (m) | Participant (%) | | | |
|---|---|---|---|---|
| | **a** | **b** | **c** | **d** |
| 3.60 | 100 | 0 | 0 | 0 |
| 3.50 | 100 | 0 | 0 | 0 |
| 3.40 | 100 | 0 | 0 | 0 |
| 3.30 | 100 | 0 | 0 | 0 |
| 3.20 | 100 | 0 | 0 | 0 |
| 3.10 | 92.3 | 7.6 | 0 | 0 |
| 3.00 | 34.6 | 65.4 | 0 | 0 |
| 2.90 | 16.2 | 83.8 | 0 | 0 |
| 2.80 | 6.9 | 65.4 | 27.7 | 0 |
| 2.70 | 0 | 0 | 33.8 | 66.2 |
| 2.60 | 0 | 0 | 0 | 100.0 |
| 2.50 | — | — | — | — |

Note: (a) I can alight without any problems; (b) I can alight but feel it is a little narrow; (c) I feel it is difficult to alight but I could do it; (d) I could not alight; (—) This setting was not attempted.

### 2.2.2. Analysis of Demographic Factors

It is noteworthy that some potential demographic factors may be related to the minimum width selected. We analyzed the correlation between characteristics of participants and selection in the minimum width (i.e., the gray cells highlighted in Table 2). One-way analysis of variance (ANOVA) was used for analysis. One-way ANOVA is frequently used to compare the means of two or more unrelated groups and the statistically significant effects of individual demographic variables on responses. Table 3 shows the results of p-value from ANOVA. The test of homogeneity of variances indicated that ANOVA was acceptable (*P* of all demographic factors > 0.05).

**Table 3.** The influence of demographic factors.

| | Gender | Age | BMI | Types of Aids | Health Condition | Driving Experience |
|---|---|---|---|---|---|---|
| Selection in minimum width | 0.289 (1.254) | 0.002 * (6.487) | 0.045 * (3.287) | 0.009 * (4.356) | 0.000 ** (8.718) | 0.000 ** (8.791) |

Note: columns show the *p* value. The numbers in parentheses are *F* values. * $p < 0.05$, ** $p < 0.001$.

Result shows age, BMI, type of aids, health condition and driving experience were correlated with the selection. However, gender had no impact on participants' selection.

## 3. Questionnaire Survey

From the experimental survey, results indicated that older participants required wider-than-standard parking spaces. The findings also initially inferred that demographic factors (e.g., age, type of aids, and driving experience) were probably related to minimum width. The second part of this project enlarged the sample and further identified the correlates between other characteristics of the older individuals and width of parking spaces through a questionnaire survey.

### 3.1. Data and Method

#### 3.1.1. Data Sources

A paper questionnaire was used in the Geriatric Service Association of Zhejiang Province in October, 2018. The Geriatric Service Association sent the questionnaire randomly to the individuals who were registered in the association. The randomization technique used in this survey was randomly selecting every tenth person in the registration list, regardless of their gender, family background or other factors [25]. The older participants included aid users and others without any aids. Considering the

influence of parking assistance systems as we did in the experimental survey, the questionnaire asked driver participants whether they were using a parking assistance system. We eliminated those from the sample who answered their cars had a parking assistance system. Participants mailed the completed questionnaire back to the Geriatric Service Association. All participants' names and other personal information were then erased by the Geriatric Service Association; therefore, all questionnaire data for the analysis were anonymous. All participants were 60 years or older.

### 3.1.2. Outcome Variables and Other Items

Participants needed to answer the question: "When you arrive at a parking lot and have, in reality, the choice to use different widths of parking spaces, do you choose: (A) one of the plentiful parking space that is easy to find even when it is narrower than standard parking spaces; (B) I search for one of the fewer parking spaces that have a standard size; (C) I keep searching for one of the very limited number of parking spaces that are wider than the standard parking space". All participants were asked to select one of the dependent variables.

In order to measure the influence of the parking spaces on parking behavior, participants were further asked to answer the question, "If you cannot find a wide enough parking space to park or alight in within fifteen minutes after you arrive at your destination, what will you do next?". The answer options included: (A) continue to look for a wider space; (B) go back home directly; and (C) other.

### 3.1.3. Sociodemographic Variables

For this analysis, sociodemographic variables included gender, age, body size (height and weight) as well as types of aids used. Gender was collated into two categories: male and female. Age was divided into three categories: 60–69, 70–79 and over 80 to categorize different ages of older adults [26]. We considered the fact that aids used by individuals might be multiple. For example, one older participant used a cane generally but he or she might use a wheelchair from time to time as well. Therefore, we asked participants to choose one of most often used aids in daily life. It is noteworthy that many wheelchair users whose disability is associated with age in this survey reported that they do not have a disabled parking permit and disabled parking spaces are not necessary for them to use. Some literature also recorded that older people do not apply for the disabled parking permit in China out of self-pride and insist on not using disabled parking spaces [27]. Therefore, we did not eliminate these wheelchair users and kept them in the analysis. The options of aids were split into three main categories: motorized/manual wheelchair, cane/others, and none. Ethnic background was not listed on the questionnaire because most residents (n = 99.15%) in Zhejiang Province are of Han nationality. It is noteworthy that we did not use body weight and height variables in the model due to the possibility of multicollinearity with body mass index (BMI), though two variables were recorded in the questionnaire. BMI was analyzed in the model as one of the health variables.

### 3.1.4. Transport Variables

We assumed all participants were using their private cars to park or had experience in alighting and boarding in a parking space. Therefore, using private cars frequently and having rich parking experience may be linked to the outcome variable [19]. We created the following transport variables: whether participants owned or had access to a car, and whether they held a driving license. In addition, participants were asked how much driving experience (in years) they had (presented as none, 0–<3, 3–<10 and ≥10) and how often they use a private car (times per week) (presented as <3, 3–<7, and ≥7).

### 3.1.5. Health-Related Variables

Previous studies have highlighted the close connection between body mass index (BMI) and health-related quality of life (HRQoL) as well as burden of disease [28–30]. Therefore, we used BMI as one of the health-related variables. The BMI was divided into four categories: underweight (<18.5), normal (18.5–<25), overweight (25–<30) and obese (≥30). Self-assessed health condition

was split into five categories: extremely bad; bad; neither good nor bad; good; and extremely good. Further questions about detailed health information or type of illness for individuals were not specified due to confidentiality.

### 3.1.6. Logistic Regression Model Analysis

Logistic regression is often used to model categorical outcome variables [30]. Therefore, we selected a logistic regression model to examine the connection between "width of space" and sociodemographic, transport or health-related variables. The detailed description and estimation method of the logistic regression model are referenced in the research contents of Luce [31]. We will not present them in detail in this paper.

Statistical analysis was performed using IBM SPSS Statistics 22. In this analysis, goodness-of-fit statistics include Max-rescaled R-Square and the Pearson test. In the Pearson test, a high *p*-value indicates that the fitted model cannot be rejected [32].

### 3.2. Results

The questionnaire survey was completed by 1179 participants. 194 cases were excluded due to missing data or because potential participants used a parking assistance system, leaving a sample of 985 (effective rate = 83.6%). In these valid samples, only 1.6% of participants (N = 16) selected the option "parking spaces can be narrower than standard parking spaces" and only 15.1% of participants (N = 149) chose "standard parking spaces are acceptable". The remaining 83.3% of participants (N = 820) stated that the widths should be wider than standard parking spaces. Concerning the question "What will you do next if you cannot find a wide enough space to alight when you arrive at your destination", about 68.1% of participants (N = 670) selected the option "go back home directly".

Before analysis, we checked the multicollinearity by using multiple linear regression. Results showed no multicollinearity among the independent variables ($p > 0.05$). The p-value of 0.952 of the Pearson test indicated the model is acceptable. Furthermore, the calculated Max-rescaled R-Square value is 0.752, which shows that the model is not very strong but a fair representation of the data.

Table 4 presents the characteristics of participants who chose the option "wider than standard parking spaces". Results showed that factors of age, weight and height, types of aids, some transport variables (e.g., driving experience; frequency of using a car) and health-related variables (BMI and health condition) were significantly related to propensity for choosing "standard parking spaces" ($p < 0.001$).

Table 5 outlines the *p*-value, the odds ratios and the 95% confidence interval for each characteristic of participants who selected ">standard space" in the model.

The results show that males were more likely to use a wider space than females. Individuals who were 60–69 years old were less likely to choose a wider space than individuals who were over 80 years old. Types of aids were proven to be significantly associated with the wider space. Individuals who were using aids chose a wider parking space.

Transport variables were correlated to the selection of a wider space. For example, participants who had a driving license and had access to a car might not need a wider parking space. Those with rich driving experience and who frequently used (or took) a car were proven to more easily get off in a smaller space.

**Table 4.** The profile of characteristics of the participants.

| Variable | Characteristic | >Standard Parking Spaces | | *p* |
|---|---|---|---|---|
| | | Count | Percentage | |
| All respondents | | 820 | 83.3% | |
| *Sociodemographic variable* | | | | |
| Gender | Male | 457 | 89.3% | 0.045 * |
| | Female | 363 | 76.7% | |
| Age | 60–69 | 449 | 80.2% | <0.001 ** |
| | 70–79 | 314 | 86.0% | |
| | ≥80 | 57 | 95.0% | |
| Types of aids mainly used | wheelchair | 184 | 98.9% | <0.001 ** |
| | Cane/others | 307 | 93.6% | |
| | None | 329 | 69.9% | |
| *Transport variable* | | | | |
| Owns or has access to a car | Yes | 316 | 77.6% | <0.001 ** |
| | No | 504 | 87.2% | |
| Has a driver's license | Yes | 271 | 71.3% | <0.001 ** |
| | No | 549 | 90.7% | |
| Driving experience (years) | None | 417 | 94.1% | <0.001 ** |
| | 0–<3 | 253 | 87.8% | |
| | 3–<10 | 78 | 61.4% | |
| | ≥10 | 72 | 56.7% | |
| Frequency of using (or taking) a car (times per week) | <3 | 496 | 92.7% | <0.001 ** |
| | 3–<7 | 241 | 81.1% | |
| | ≥7 | 83 | 54.2% | |
| *Health-related variable* | | | | |
| BMI | <18.5 (underweight) | 215 | 89.2% | <0.001 ** |
| | 18.5–<25 (normal) | 306 | 76.1% | |
| | 25–<30 (overweight) | 183 | 84.3% | |
| | ≥30 (obese) | 116 | 92.8% | |
| Health condition | Extremely good | 18 | 25.7% | <0.001 ** |
| | Good | 123 | 66.1% | |
| | Neither good nor bad | 334 | 89.5% | |
| | Bad | 253 | 95.8% | |
| | Extremely bad | 92 | 100.0% | |

Note: * $p < 0.05$, ** $p < 0.001$.

In addition, BMI and self-reported health conditions were also significantly associated with the selection of a wider space. Participants whose BMI was between 18.5–25 were less likely to choose a wider space than individuals with BMI > 30. Individuals who reported they were in extremely good health condition were less likely to park in a wider parking space than those who claimed they were in extremely bad health condition.

Table 5. Logistic regression analysis of ">standard parking space".

| Variable | Characteristic | Odds Ratio | 95% C.I | *p* |
|---|---|---|---|---|
| Gender | Male | 2.518 | 1.771–3.579 | 0.000 ** |
| | Female (ref cat) | | | |
| Age | 60–69 | 0.213 | 0.095–0.371 | 0.000 ** |
| | 70–79 | 0.324 | 0.212–0.435 | 0.000 ** |
| | ≥80 (ref cat) | | | |
| Types of aids mainly used | wheelchair | 39.682 | 31.322–47.042 | 0.000 ** |
| | Cane/others | 6.310 | 3.145–9.520 | 0.000 ** |
| | None (ref cat) | | | |
| Owns or has access to a car | Yes | 0.510 | 0.364–0.715 | 0.000 ** |
| | No (ref cat) | | | |
| Has a driver's license | Yes | 0.454 | 0.410–0.502 | 0.000 ** |
| | No (ref cat) | | | |
| Driving experience (years) | None | 11.793 | 9.589–13.003 | 0.036 * |
| | 0–<3 | 5.522 | 4.387–6.664 | 0.000 ** |
| | 3–<10 | 1.216 | 0.725–1.797 | 0.215 |
| | ≥10 (ref cat) | | | |
| Frequency of using (or taking) a car (times per week) | <3 | 10.726 | 6.802–16.913 | 0.043 * |
| | 3–<7 | 3.630 | 2.359–5.585 | 0.048 * |
| | ≥7 (ref cat) | | | |
| BMI | <18.5 | 0.642 | 0.581–0.699 | 0.000 ** |
| | 18.5–<25 | 0.347 | 0.237–0.474 | 0.007 * |
| | 25–<30 | 0.518 | 0.424–0.586 | 0.000 ** |
| | ≥30 (ref cat) | | | |
| Health condition | Extremely good | 0.257 | 0.173–0.383 | 0.000 ** |
| | Good | 0.661 | 0.597–0.733 | 0.000 ** |
| | Neither good nor bad | 0.895 | 0.865–0.927 | 0.000 ** |
| | Bad | 0.958 | 0.935–0.983 | 0.017 * |
| | Extremely bad (ref cat) | | | |

Note: * $p < 0.05$, ** $p < 0.001$, ref cat = reference category

## 4. Discussion

Based on an experiment and a questionnaire survey, this study analyzed the minimum width of parking spaces for the elderly and the factors that influence the selection of wider or narrower spaces. Our findings have confirmed that most older participants required a wider-than-standard parking space. These findings are broadly consistent with some previous research in South Korea and Japan [14,19,33]. In addition, when considering older individuals who are using a cane or those without any aids, we found that they accepted a parking width of 2.70 m.

The findings also indicated that age, types of aids, driving experience, BMI and health condition were main factors that might influence selection behavior of parking spaces for the elderly. (I) Increasing age was one of factors. Older individuals generally have fewer opportunities to travel by car and park than younger individuals owing to their health conditions. Moreover, physical ability and concentration skills generally worsen with increasing age, which might cause difficulty of parking in a relatively narrower space. (II) Aids were tightly related to selection of wider or narrower space. Participants who were using aids seem to be associated with wider parking spaces, as most participants required a wider space. (III) Having rich experience in driving a car was proven as an influential factor as well. A plausible reason for this phenomenon may be that those individuals have more experience to park as they own a private car; therefore, they are more likely to have higher parking skills than others who do not use private cars. This viewpoint was also confirmed by the results from the

questionnaire and experiment, which reported most participants with less driving experience needed a wider space. (IV) BMI was shown as a powerful predictor. (V) Health condition shows strong evidence to be associated with "wider parking spaces". This result makes sense because the sensorimotor and cognitive functions commonly become worse if an individual is not in a good health condition.

The gender differences did not demonstrate a significant connection with "wider parking spaces" in our study. Nevertheless, the results from the questionnaire showed the percentage of male participants preferring larger spaces is greater than females, implying selection might be influenced by gender differences. The gender differences may attribute to physical, psychological, and other factors. In addition, males are different from females in regard to functional capacities, perceptions of safety and social norms [34]. For instance, the gap in the need of wider spaces between older men and older women may be because older women had more chances to go driving for shopping than older males and, therefore, may have more chances to park in standard spaces than older men in China and Japan [35]. However, this hypothesis may not fit in other countries in the West owing to respectively differing lifestyles of males and females there compared with those in East Asian countries. Another possible explanation is that body size of males is relatively bigger than females [36], suggesting male individuals need more space to alight, which is shown as the results of experimental and questionnaire surveys.

It is widely known that being mobile and using transport are considered to increase individuals' independence during later life while aging [37–42]. Availability of parking facilities has also been shown to affect travel behavior [43–47]. Some researchers have pointed out that reducing car access and car parking facilities will cause a reduction of outdoor activities and trips for the elderly, in particular, in small cities and rural areas [48,49]. Our finding is consistent with previous literature and has indicated wide enough parking facilities would influence the elderly to travel and participate in social activities (about 68.1% of participants would stop participating in social activities and go back home directly if they could not find a wide enough parking space at their destination).

Adding wider parking spaces for the elderly is one method to ensure older adults can alight easily; however, it is very difficult to set many wider spaces in urban cities in China. Moreover, allowing all elderly to use disabled parking spaces may bring more controversies as contention about accepting older people using disabled parking spaces has erupted in Japan [14,50]. Lu et al. [18] recorded a public symposium held in Japan about the rationale of expanding the amount of disabled parking spaces. Twelve older people who were using canes and walkers agreed that current disabled parking spaces were quite big for their usage. Furthermore, the wheelchair users pointed out that these older individuals should not be entitled to use disabled parking spaces due to the fact that disabled parking resources are very limited and most elderly do not actually need such parking spaces. The root issue is that the limited number of reserved spaces (N = 2% of total parking spaces) is not sufficient to provide parking resources to all older adults in China [23]. The authorities do not have a good solution thus far.

## 4.1. New Concept of a Reconfigured Parking System

The current parking system can be reconfigured to make sure the elderly park safely and smoothly based on our findings. For example, we can add these older wheelchair users to use current disabled parking spaces (note: a disabled parking space is 3.60–3.70 m-wide in China) and create another relatively narrow type of space (e.g., 2.80 m-wide) for others in order for all the elderly to be able to board more easily. In particular, these 2.80 m-wide parking spaces for the elderly may be very important in the direct surroundings of hospitals, health care facilities, and other offices that older people frequently visit. However, more practical tests of the amount of demand for such parking spaces are needed.

It is noteworthy that a vehicle parking assistance system that includes a sensor system would be very helpful for older drivers when they are parking. One survey in Japan [51] has presented that eighteen elderly drivers drove and parked their vehicles comparatively easily when using a vehicle parking assistance system when comparing to those without an assistance system. Many older drivers

will benefit from the effect of the large-scale use of such systems in cars and should find it much easier to park. However, not every elderly driver is able to install a vehicle parking assistance system due to high costs. Local governments in the future may need to provide subsidies to install vehicle parking assistance systems for older drivers.

In addition, a drop-off zone may be very useful and important for older passengers from both private cars and public transport (e.g., bus), because it connects transport (both public and private patterns) and destinations well during boarding and alighting processes. Bascom and Christensen [52] found that many older people used a car to travel or participate in social activities with assistance from their family or friends. Therefore, older car passengers can first be dropped off at the drop-off zone with the help from their companions or volunteers. Then these older passengers will be picked up after the driver parks in a regular parking space. For the bus passengers, bus stops can be set at (or nearby) the drop-off zone, which is closest to a destination. Furthermore, some literature stated that older passengers had reported that they could not step easily onto the pavement of the bus stop (or had to jump off the bus) [10]. The height of a drop-off zone for buses may be designed to be aligned with the height of the bus exit because older passengers have trouble with the alighting process. A well-trained bus driver or volunteer can help older passengers board or alight the bus. This idea should be useful because it can be predicted that in a sustainable society more and more elderly people will use buses (or other public transport modes) to travel with the improvement of infrastructure and public transport [9]. Moreover, older drivers who have severe trouble in walking (e.g., wheelchair users) can alight at disabled parking spaces. Older drivers who have mild trouble walking can use 2.80 m-wide parking spaces (the narrow type). Healthy older people can park in 2.80 m-wide parking spaces or standard parking spaces. In addition, the distance between the destination and the actual location of the parking space also influences the satisfaction with parking facilities. People are, on average, willing to accept approximately 100 m between destination and destination parking [46]. Therefore, the parking space and drop-off zone for the elderly should be set as close to the destination as possible. Figure 2 shows the complete concept of a reconfigured parking system for older adults with different categories. These suggestions for adjustments should provoke interest in community leaders who consider improving parking spaces for the elderly in resource-limited cities, though enforcement in many areas is still very challenging. Nevertheless, our study has proposed a concept for how to provide more wider parking options for the elderly and help the elderly alight easily and safely in both private cars and buses.

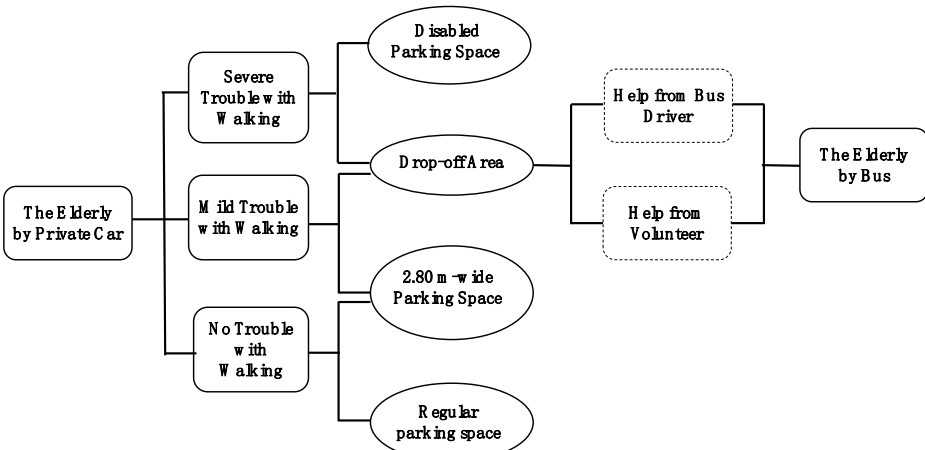

**Figure 2.** Concept of a reconfigured parking system for the elderly from private cars and buses.

*4.2. Limitations*

The study limitations include that the sample is from only one metropolitan area. We will enlarge sample populations and continue experimental surveys in other areas in China, as well as in other countries. The weather conditions in the experimental survey were good. We may carry out the field

experiment in some inclement weather conditions to test the requirement of appropriate width in different conditions. In addition, we mentioned that a vehicle parking assistance system might be very useful for parking for older drivers. In the future, we will test the effect of a vehicle parking assistance system's effect on the minimum width of parking spaces for elderly drivers.

The other limitation that we did not consider is the trade-off between benefits and costs in the outcome variable in the paper questionnaire. We will set outcome variable choices related to preference and real-world choice behavior in the future. Nevertheless, our findings have indicated some important influencing factors affecting selection, which will be helpful and valuable in the further research.

In addition, we cannot confirm the accuracy of self-reported data (e.g., self-reporting if width of parking spaces was reasonable and self-reported of health condition) in the two surveys [53,54]. For example, some participants are likely to select a wider parking space option because it may be much easier for them to park. In addition, the accuracy of self-reported health condition was difficult to estimate due to confidentiality [55,56]. In future research, we should add a list of diseases and illnesses to specify which type of health issues those who self-report a bad health condition suffer.

## 5. Conclusions

We examined the minimum width of space required by older adults and identified the correlates of selection of wider or narrower spaces by the elderly in China by two surveys. The initial one has presented that older participants required wider than standard parking spaces. A width of 2.80 m was adequate for most older adults. The second one showed that age, types of aids, driving experience, BMI, and health condition were main potential factors that influenced the selection of wider or narrower parking spaces. According to the findings from two surveys, we presented a new parking system concept that would encourage travel and enhance freedom for the elderly.

**Author Contributions:** Conceptualization, W.L. and C.Z.; data curation, C.Z. and X.N.; formal analysis, W.L. and C.Z. and X.N.; investigation, W.L. and C.Z. and X.N.; methodology, W.L. and X.N.; software, W.L. and C.Z.; validation, W.L. and C.Z. and X.N.; writing—original draft, W.L. and H.L.; writing—review and editing, W.L. and H.L. All authors have read and agreed to the published version of the manuscript.

**Funding:** This research was funded by the Natural Science Foundation of China, grant number 51508512 and the Fundamental Research Funds of Zhejiang Sci-Tech University, grant number 2019Q055.

**Conflicts of Interest:** The authors declare no conflict of interest.

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
