# Peer review of "Do the Elderly Need Wider Parking Spaces? Evidence from Experimental and Questionnaire Surveys"

_sustainability, doi:10.3390/su12093800_

Round 1

Reviewer 1 Report

  Current topic, interesting, worth researching and analyzing.   The title includes a reference to sustainable society, which has not been broadly defined in the content, it does not indicate what it is, what makes it stand out. It is worth supplementing or removing from the title.   Well defined variables in research. Obtained test results presented legibly and logically, correct conclusions. Various and correct research methods and techniques used.   It was interesting for the reviewer to highlight the cultural differences between the Chinese, Japanese, other Asian and European societies in terms of determinants of factors affecting parking experience for men and women.   In 4.1 it is worth adding, as a possible solution - the introduction of wider parking spaces for the elderly (2.8m) in front of selected objects, such as offices, health care facilities, to which older people move more often and practical testing of the amount of demand for such parking spaces .

Reviewer 2 Report

The paper describes an interesting study in two steps in a consistent way. I have a few issues that to my opinion needs to be addressed in an update of the paper. 

First, today many (new) cars have park assist systems, helping the driver to very precisely park. a. In your selection of respondents for the paper questionnaire, did you ask for whether people use a park assist system for parking and did you eliminate these from the sample?. It makes a big difference in perception and experience of complexity of the parking process: fully manual or supported by a system. b. In relation to section 4.1: can you add considerations regarding the effect of large scale use of such systems in cars? E.g. Perhaps giving all elderly drivers the opportunity to install such a system in their cars against low costs (subsidy?), would already solve the problems observed in the paper. 

Second. In reference to 3.1.2 (outcome variables), lines 191-192. It is mentioned that people were asked to select a particular type of parking lot. This question is presented unframed and consequently, it is unclear what was asked. Is it: when you have in reality the choice between A,B and C, which one do you choose?. Or: in case you can park in a parking lot that is smaller than your preference, do you start seeking for another one or do you accept it? Many other possible phrasings of the question can be thought of. Please make clear. Perhaps move lines 237-239 to here, because these seem to give a better impression of the kind of questioning. 

Third. When one asks what preference somebody has, it is very likely that the largest / best option (more space) is preferred. This doesn't come to a surprise. However, in practice normally a choice comes with a price, and a trade-off between benefits and costs have to be made. Why didn't you ask to measure choices in terms of "suppose you can choice A, B or C, and that the costs for A (small parking lot) are X-20%, for B (normal parking lot) are X, and for C (wider than normal) are X+20%, what would you choose?". This would give a better indication of the relationship between preference and real world choice behaviour ("I accept the normal parking lot, because the extra price of the wider one does not weight against the more easy parking"). So explain why you haven't chosen to force respondents to make a trade-off, but instead only measure their preference, because this might bias the resulting picture significantly. 

Fourth. Also in relation to the previous two issues: you haven't made clear in the paper whether drivers actually do not use the car, or avoid using the car for certain destinations, to avoid parking problems. So when you conclude (line 292), that you have proven that the standard parking space is undersized for elderly, this is strictly not true. The study shows that elderly people in general have more preference for larger spaced parking lots (because the have more difficulty getting in/out he car on smaller parking lots), but you have not provided any evidence for the implicit assumption that too few wider parking places in reality limits car use of the elderly . So strictly speaking, the proposal to change the parking system because it would encourage the elder population to more participate in society (line 328-329), cannot be based on the presented analysis. I want the authors to add evidence for the influence on car use and/or parking behaviour (e.g. parking further away from the destination), or at least address these issues in the discussion section. 

Fifth, I think the writing in English needs to be carefully checked. It is not at the level yet for publication. In addition a few textual things, (too many small textual details not included):

  • line 74: "the fact" should be "the hypothesis"
  • line 114: reference is 1992, which is for this information very old. More recent information? 
  • line 128: ".... try to ... expect"; I don't understand this sentence.
  • section 3.2 describes results. Unfortunately, it tends to repeat figures that are also mentioned in the table and that makes the text less readable. Is it possible to be more selective and focus on the interpretation / the narrative
  • line 368: "the sample of the sample" ? 

Reviewer 3 Report

Dear Authors

The issue raised in the article is very interesting but controversial from the point of view of the sustainable development concept. The issue of communication exclusion of indyvidual groups of society is very important and current. The important questions are:

  • Do we want older people to drive their own vehicles?
  • Is it and will it be safe for them and for other road users?
  • Will the introduction of facilities in parking lots mean that instead of using public transport they will choose individual vehicles?

Public transport is becoming more and more friendly for people with disabilities, both infrastructure and vehicles. If the elderly person is disabled, they are entitled to a parking space for the disabled person. Why more parking spaces of this type should be design? Don't you think this is encouraging older people to buy and use private vehicles?

According to the reviewer, scientists are conducting research not only for science itself but, above all, for the economy, socjety and environment. What are the benefits of promoting private transport?

Other remarks:

  • Row 111 – What did the vehicle colors mean? Is this information really needed?
  • Row 124 – „which is how most drivers usually park in China” – How do You know it? Are there any researches about it available? If yes, source is needed.
  • Figure 1 – it should be a technical drawing, not a photo
  • Figure 2 – it’s too trival. It’s unacceptable in academic research
  • Table 1 - Driving experience (years) – set boundary values are incorrect: a person with 3 years of experience could choose 2 sets: 1st and 2nd
  • Table 5 – the same remark: <3-5>, <5-10>
  • Rows 198-200 – „For example, one older participant used a cane generally but he/she may use a wheelchair from time to time as well. Therefore, we asked participants to choose one of most often used aids in daily life” – Isn’t a wheelchair user a disabled person? Shouldn't such a person use a parking place for disabled people?
  • Rows 208-210 – „We assumed all participants were using their private cars to park or had experience of alighting and boarding parking space by cars. Therefore, using private cars frequently and having rich parking experience may be linked to the outcome variable” – but it’s but it is contrary to the assumptions of sustainable development
  • Too few References – only 38
  • Literature review is covered mostly by Asian literature. European and American literature has been used in a very limited number, although it is very accessible.
  • Conclusions are obvious
  • Row 386 – only with indywidual vehicle

Round 2

Reviewer 2 Report

I thank the authors for the provided improvements, covering most of my questions and remarks. 

Some remaining remarks:

  • In line 96-97 you mention "which includes a drop-off zone for buses' elderly passengers". This link to buses passengers is strange, because the study is about width of parking spaces. I suggest to delete this part of the sentence. 
  • line 105 says age limit 76, line 106 mentions 75. 
  • line 158 mentions that BMI is calculated using weight and length. Unless you calculate BMI in a way that deviates from the standard, this sentence is not needed. It is common knowledge that these 2 variables are the basis for calculation. The calculation of BMI is also mentioned in line 244-245
  • line 1980199: change "we eliminated the sample" in : we eliminated those from the sample who ..."
  • I appreciate the formulation of the options in line 205-208, but the formulation themselves are odd. You say: make a choice out of the following 3 options. My suggestions is to formulate them as: "A.one of the plenty parking space that is easy to find even when it is narrower than standard parking spaces. B. I search for one of the fewer parking spaces that has a standard size. C. I keep searching for one of the very limited number of parking spaces that is wider than the standard parking space".  And eliminate "which one will you choose?"in line 208 (because it doubles with line 205). 
  • line 2982: change "Results presented" in "The results show that..."
  • line 298: replace 'confirms' by "analyzed the" or "explored the..."
  • line 314: replace "satisfied " by 'confirmed'
  • line 320: delete "greatly"(connection is yes or no significant) 
  • line 324: replace "are dissimilar" by "are different from"
  • line 358 "in hospitals"-> "in the direct surroundings of" 
  • from line 368 and further: I find it strange why you start discussing drop-offs zones for bus passengers. This is not the focus of the research and you haven't made an analysis of possible problems. What is it different from a bus stop? And why mix it with your suggestions for improving the parking system. For me, it makes no sense to add these thoughts and it is not based on the described research. 
  • A final remark: I still think the English language can be improved. In particular in many places "the" is missing. 

Reviewer 3 Report

Thanks to the authors for the corrected version of the manuscript. You've done a lot of good work. Congratulations.

Author Response

Thank you for your comments and suggestions which help to improve this paper a lot.